# Epizootic Hemorrhagic Disease Virus: Current Knowledge and Emerging Perspectives

**DOI:** 10.3390/microorganisms11051339

**Published:** 2023-05-19

**Authors:** Luis Jiménez-Cabello, Sergio Utrilla-Trigo, Gema Lorenzo, Javier Ortego, Eva Calvo-Pinilla

**Affiliations:** Centro de Investigación en Sanidad Animal (CISA-INIA/CSIC), Valdeolmos, 28130 Madrid, Spain; lfj.cabello@inia.csic.es (L.J.-C.); sergio.utrilla@inia.csic.es (S.U.-T.); lorenzo.gema@inia.csic.es (G.L.); ortego@inia.csic.es (J.O.)

**Keywords:** EHDV, epidemiology, vaccines, white-tailed deer, cattle

## Abstract

Epizootic Hemorrhagic Disease (EHD) of ruminants is a viral pathology that has significant welfare, social, and economic implications. The causative agent, epizootic hemorrhagic disease virus (EHDV), belongs to the *Orbivirus* genus and leads to significant regional disease outbreaks among livestock and wildlife in North America, Asia, Africa, and Oceania, causing significant morbidity and mortality. During the past decade, this viral disease has become a real threat for countries of the Mediterranean basin, with the recent occurrence of several important outbreaks in livestock. Moreover, the European Union registered the first cases of EHDV ever detected within its territory. Competent vectors involved in viral transmission, *Culicoides* midges, are expanding its distribution, conceivably due to global climate change. Therefore, livestock and wild ruminants around the globe are at risk for this serious disease. This review provides an overview of current knowledge about EHDV, including changes of distribution and virulence, an examination of different animal models of disease, and a discussion about potential treatments to control the disease.

## 1. Introduction

The appearance and spread of outbreaks caused by arboviruses that affect both wild and domestic ruminants carry a high risk of generating important direct and indirect economic losses. Therefore, control of diseases caused by arboviruses is essential to ensure the welfare of livestock, as well as to prevent possible detrimental effects on local, regional, and national economies. In this regard, implementation of vaccination campaigns against this type of pathogen has successful, with a great impact on animal health. However, a variety of anthropogenic factors, climate change, and growing global trade increase the risk of appearance of viral diseases, some of them transmitted by arthropod insects, in non-endemic territories. The most recent example of this is recent landing in Europe of an orbivirus previously unknown on this continent, the epizootic hemorrhagic disease virus (EHDV).

Epizootic hemorrhagic disease (EHD) is an arthropod-transmitted viral illness of wild and domestic species of the suborder *Ruminantia*, included in the list of notifiable diseases of the World Organization for Animal Health (WOAH) since 2008. The causative agent of EHD is EHDV. This virus belongs to the genus *Orbivirus* within the family *Sedoreoviridae*, similar to bluetongue virus (BTV) and African horse sickness virus (AHSV). EHDV and BTV share a variety of domestic and wild ruminant hosts, including sheep, white-tailed deer (WTD), and cattle, although susceptibility to clinical disease associated with these viral infections varies greatly among host species, individuals, and viral serotype. For BTV, more than 29 serotypes have been described, while just 7 serotypes of EHDV are currently described. Here, we review fundamental aspects of EHDV and the epidemiology of this emerging and re-emerging viral disease. Moreover, we offer an outline of the pathology induced during infection of ruminant hosts as well as a description of the animal experimental models used for the study of EHD. Finally, we review the current state of vaccines against this veterinary relevant orbivirus and the prospects and challenges of next-generation vaccines.

## 2. EHDV, the Etiological Agent of EHD

Similar to members of the *Orbivirus* genus, EHDV non-enveloped virions present a structure characterized by an icosahedral capsid, which is divided into three consecutive protein layers: the inner and intermediate layers (core) and an outer capsid [1] (Figure 1). The genome is located inside the core particle and comprises ten linear double-strand RNA segments that encode for seven structural (VP1–VP7) and at least four non-structural proteins (NS1, NS2, NS3/NS3A, NS4, and probably the putative NS5 as for BTV [2]) [1,3,4]. As for the prototype BTV, the outer protein layer is made of 60 trimers of VP2, the most exposed virion protein, and 120 trimers of VP5 [5]. The inner capsid is made of the subcore, formed by VP3, and the intermediate layer, constituted by VP7, along with three minor structural proteins with enzymatic activities: VP1 (RNA-dependent RNA polymerase), VP4 (capping enzyme), and VP6 (helicase) [4,6,7,8,9].

Outer capsid proteins VP2 and VP5, encoded by segments 2 and 6, respectively, are the most variable proteins among EHDV serotypes, especially VP2, probably due to great selective pressure [10]. Similar to BTV, VP2 and VP5 accomplish key roles during the early stages of infection, being involved in virus attachment and virus entry into host cells [11]. VP2 is also mainly responsible for the induction of virus neutralizing antibodies (nAbs), thereby defining virus serotype based on cross-neutralization assays and supported by extensive phylogenetic studies [10]. Rapid, sensitive, and specific molecular typing assays have been developed by identification of segment 2 nucleotide regions unique to each EHDV serotype [12,13].

VP1 (segment 1), VP4 (segment 4), and VP6 (segment 9), involved in genome replication, are also highly conserved proteins, showing at least 85% of amino acid sequence identities when eastern and western strains were considered separately [14]. Identification of conserved nucleotide regions in segment 9 allowed development of a highly sensitive EHDV pan-reactive TaqMan real-time RT-PCR assay for diagnosis and genome detection in tissues [12].

VP7 (encoded by segment 7) shows a highly conserved amino acid sequence (more than 90% sequence identity) among EHDV serotypes [14,15]. VP7 is the immunodominant serogroup-specific protein and is used in serogroup specific enzyme-linked immunosorbent assays (ELISAs) for disease diagnosis [16]. Recently, researchers have shown the usefulness of purified and unpurified baculovirus-expressed VP7 for development of competitive enzyme-linked immunosorbent assays [17,18].

Non-structural proteins, found in infected cells but not in virus particles [19], play supportive roles in crucial viral processes such as genome packaging, intracellular transport, capsid assembly, virus release, and control of the immune response. NS1, the most expressed protein during orbivirus replication in infected cells and positive regulator of viral protein synthesis, forms tubules in the cytoplasm that are involved in viral transport within infected cells and have a role in cellular pathogenicity [20,21,22]. NS1 has been described as an almost identical protein among EHDV serotypes (more than 90% sequence homology) [23]. The highly conserved protein sequence of NS1 is thought to be important for tubule formation and function. In this sense, the conservation of 16 cysteine residues in the protein sequence, including two residues at positions 336 and 339, could be of fundamental importance for NS1 formation as it occurs in the case of BTV [23]. The highly conserved nucleotide sequence of NS1 has also enabled development of rapid molecular tools for diagnosis of EHDV [24], even in samples of aged bone marrow for up to 12 weeks after animal death [25]

NS2 (segment 8) acts as ATPase, having a role in RNA packaging and translation [26]. This phosphoprotein is the major component of viral inclusion bodies (VIB), acting as virus assembly sites [27,28]. In terms of amino acid identity, NS2 varies more than NS1. The sequence identity seen in NS2 among all strains varies around 79.7% [23]. Authors also reported the conservation of a domain at the N-terminus of NS2, between amino acids 75–83. This domain was shown to be important for binding single-stranded RNA and formation of VIBs during EHDV infection [26]. Differences at the sequence of this domain between western and eastern strains could explain differential pathogenicity in cattle [23].

NS3 (segment 10) is important in virus release from infected cells, being expressed at greater levels. NS3, which possess transmembrane domains [29], facilitates virus egress via budding, rather than lysis, in Culicoides cells [30]. NS3 and NS3A are closely related proteins, both translated from sequential ORFs of segment 10 [31]. The sequence identity mean of NS3 is 89.7% at the amino acid level between eastern and western viruses [23].

For NS4 (encoded by segment 9), which is a virulence factor in the case of BTV [32,33], no sequence homology analysis between EHDV serotypes has been conducted so far.

## 3. Changes in EHDV Epidemiology: Influence of Global Warming

To date, seven serotypes of EHDV have been identified, named as 1–2 and 4–8, designated based on extensive phylogenetic studies, sequencing data, and cross-neutralization assays [10]. Genetic analyses demonstrated that previously identified serotype 3 [34] (Nigerian strain Ib Ar 22619) was serotype 1 [10]. EHDV has been isolated in North and South America, Africa, Asia, the Middle East, and Oceania (Figure 2a). To date, it is endemic in parts of North America, Australia, and certain countries of Asia and Africa [35]. EHDV was first detected in the USA in 1955 when WTD were severely affected showing high mortality [36]. Among the seven serotypes proposed, EHDV-1, 2, and 6 have been reported to be present in North America, where WTD is the most severely affected host and the scale of individual outbreaks increased with time [37,38,39]. In Australia, six out of seven serotypes (EHDV-1, -2, -5, -6, -7, and -8) have been detected over the years [35,40]. Globally, the presence of EHDV has been noted in Japan (serotypes 2 and 7, and serotypes 5 and 6 recently isolated from *Culicoides* insect vectors), China (serotypes 1, 5, 6, 7, 8), Morocco (serotype 6), Algeria (serotype 6), Libya (serotype 6), Turkey (serotype 6), Tunisia (serotypes 6 and 8), Egypt (serotype 1), Oman (serotype 2 and 6), Sudan (serotype 5 and 6), Nigeria (serotypes 1 and 4), the island of Mayotte (serotype 6), French Guiana (serotypes 6 and 7), Ecuador (serotype 1), Trinidad (serotype 6) and Israel (serotypes 1, 6 and 7) [41,42,43,44,45,46,47,48,49,50,51,52,53,54,55,56,57,58,59]. Genome detection or serological evidences also indicate the presence of EHDV (unknown serotypes) in Indonesia, Taiwan, Zimbabwe, Kenya, Kazakhstan, and Brazil [60,61,62,63,64]. Nucleotide sequencing and neutralization tests suggest novel strains of EHDV identified in South Africa and China as new putative serotypes [65,66]. Traditionally, this virus was thought to cause severe disease only in deer and mild illness in cattle, except for the outbreaks of EHDV-2 (traditionally known as Ibaraki virus) that have generated economic losses in Japan since the mid-20th century. The first outbreak in the country in 1959 resulted in 4000 deaths among cattle [67]. Moreover, further infection events in Reunion Island and USA demonstrated the capacity of serotypes 6 and 7 to cause severe disease in cattle as well [68,69,70]. Recent introductions of novel serotypes 6 and 7 in Israel and China also demonstrated increased severity and prevalence among bovine populations [51,71]. Importantly, field strains causing EHDV-outbreaks in Israel were likely to be abortifacient in cattle [72]. After spreading countrywide, endemization of EHD in the country is quite probable [73]. The unique assessment of economic losses caused by EHDV correspond to the EHDV-7 outbreak that occurred in Israel during the fall of 2006, with an estimated loss ranging from USD 1,591,000 to USD 3,391,000, mainly associated with a reduction in milk production [74].

Traditionally, EHDV distribution has been enclosed in temperate and tropical climates that support vector populations. In Canada, where EHDV was limited to southwestern areas [75], detection of EHDV-2 genomes and serologic evidence in deer and cattle strongly suggest that there is a progression of the virus to northern regions of America where it was not present before [76,77]. This is likely to be related to global warming. These climatic consequences were gradually visible through the EHDV outbreaks registered in the USA during recent decades and patently apparent during the major outbreak of EHDV the occurred during the summer of 2012 that coincided with severe drought and abnormally high temperatures [78]. There were large losses of WTD, and a significant amount of clinical disease in cattle in the Midwest and northern Plains [39]. During the past decade, there have been important changes in the pattern of disease and distribution of EHDV: first, related to the aforementioned growing severity of the disease among bovine populations; second, regarding the emergence of outbreaks in the Mediterranean Basin in former EHDV-free territories [79]. Outbreaks during 2020 in Turkey and North Africa were originated by serotype 6, while serotype 8 was the causative virus of several outbreaks in cattle in Tunisia during 2021 [50]. This was the first evidence of EHDV-8 circulation since 1982, when this serotype was isolated in Australia. In Europe, there was no evidence of the presence of EHDV as far as we know. However, it emerged on the continent for the first time in October 2022. After BTV-like clinical signs were noticed in some animals, EHDV was identified as the causative agent of cattle disease outbreaks in Sicily and southwestern Sardinia (Figure 2b). Subsequently, it was confirmed that these outbreaks were caused by EHDV-8, an identical strain to the one circulating in Tunisia between 2021 and 2022, pointing to North Africa as the direct origin [80]. Consecutively, EHDV outbreaks were detected in southern Spain, with EHDV serotype 8 confirmed and notified to the WOAH as the causative agent (Figure 2b).

Epidemiologic changes of arboviral diseases are also related to genetic evolution of their causative viruses. Recent works have established a distinction between EHDV strains based on their eastern or western origin, which could possibly be influenced by both regional and genetic factors [14,23]. Genetic variation present among EHDV strains derives from mechanisms of recombination, gene duplication, and point mutation [10,23]. In this sense, an African ancestry of US and Australian strains has been suggested by the identification of a recombination event in segment 8 of EHDV. Gene reassortment is another major force that characterizes EHDV evolutive strength to maintain viral fitness. In the USA, a novel EHDV-6 reassortant strain (Indiana strain) was identified as the causative virus of an outbreak in WTD during September and October 2006. At first, researchers identified a reassortment between the North American topotype of EHDV-2 and an exotic strain of EHDV-6 [70]. Thereafter, this exotic strain was confirmed to have an Australian origin [81]. Subsequently, an EHDV-6 strain isolated in Trinidad was recognized to be a product of reassortment between Australian EHDV-6 and EHDV-2, and probably USA-circulating EHDV-1. Moreover, authors pointed out this virus as a minor parent of the EHDV-6 Indiana strain [47]. Another EHDV-6 strain isolated in the USA had an Australian origin [82]. Multiple reassortment events as well as preferential reassortment have been observed between USA-circulating EHDV strains of serotypes 1, 2 and 6 [83,84]. All of these exemplify the plasticity of EHDV to survive and adapt to diversity of environmental niches, illustrating the unpredictable epidemiological features of this disease.

### Role of Culicoides Insect Vector in EHDV Spread

EHD is a vector-borne viral disease [85], so that its distribution is inherently limited to the distribution of competent *Culicoides* vectors. EHDV vectors, adult female *Culicoides*, are small biting midges belonging to the family *Ceratopogonidae* (Diptera order). Available data in the literature suggest that the species of *Culicoides* involved in EHDV transmission are likely to be similar to those that transmit BTV [86]. *Culicoides* species that are known to be implicated in EHDV transmission include *C. imicola*, *C. sonorensis*, *C. obsoletus*, *C. brevitarsis*, *C. mohave*, and *C. oxystoma*, among others [87,88,89,90]. EHDV infections are typically seasonal, occurring when vector insect populations are most abundant usually from mid-summer to late autumn [88]. However, global warming is likely to influence factors related to the incubation period and the distribution of competent insect vectors.

Many human and environmental factors clearly affect the dynamics of vector-borne diseases. Global arbovirus distribution has expanded in recent decades as a consequence of many anthropogenic factors, including commercial movement as well as long-distance travel. In addition, global warming is directly influenced by human activities and significantly affects the activity of arthropods [91,92]. It is a worrying fact that rate of warming since 1981 is more than twice as fast, 0.32 °F (0.18 °C) per decade [93], with the mean global temperature increasing more than 1 °C above preindustrial levels [92]. The ability of arthropods to transmit any virus is influenced by environmental factors such as temperature. There is evidence that higher temperatures would decrease the extrinsic incubation period of viruses and this would benefit the transmission of EHDV for longer intervals of time [94]. As an example, studies related to vector competence of *C. sonorensis* revealed that infection rate of this species with two orbiviruses was much greater at >20 °C compared with 15 °C [95]. Modeling studies of *C. imicola* future distribution suggested that its habitat suitability is likely to expand to higher latitudes in the northern hemisphere, such as Norway, Sweden, Finland, and the Kola Peninsula [96]. 

Like other arboviruses, the distribution of many orbiviruses has been increasing in recent times. As a precedent, AHSV spread in 2020 without any warning from sub-Saharan Africa towards Southeast Asia [97]. The related BTV shares with EHDV many susceptible ruminant hosts and transmission species of *Culicoides*. BTV has spread across central and north Europe since appearing on the continent in 2006, leading to continuous outbreaks and great economic losses. With this previous scenario, it is probable that EHDV as vector-borne virus could have spread among territories of the EU, especially now that its presence in Spain and Italy has been confirmed. Another factor to take into consideration is the potential distribution of *Culicoides* midges by the wind. EHDV vectors can be passively dispersed over long distances by prevailing winds [98] and this could lead to rapid spread of EHDV. For BTV, modeling studies have established the positive relationship between wind density and viral case density [99]. In the case of EHDV, winds were proposed as the major contributor to long and medium distance EHDV distribution during the Israeli outbreak in 2006. The recent arrival of EHDV to the southern territories of the EU might have originated with wind flows from Northern Africa, without dismissing other anthropogenic factors. 

EHDV can be transmitted very rapidly through several species of *Culicoides* insects present in Europe [100,101]. Coupled with global climate change, this increases the risk of introduction of the pathogen in new areas even including the central parts and north of the continent, where introduction of WTD has been carried out along with the presence of susceptible European cervids and livestock species. 

## 4. Disease and Pathology

EHD was firstly described in WTD in New Jersey in 1955 [36]. WTD (*Odocoileus virginianus*) are especially susceptible to severe disease. In North America, where EHDV is a major cause of mortality in WTD [102], periodic outbreaks result in significant mortality, but susceptibility to disease has been shown to vary between subspecies [103]. Clinical disease due to EHDV has been also reported in mule (black-tailed) deer, bighorn, yak, elk, brocket deer, and pronghorn antelope [63,102,104,105,106,107]. While sheep are highly susceptible to BTV with significant mortality, these are often resistant to EHDV-induced disease. However, it is important to consider animals with subclinical infections as reservoirs of infection that can amplify virus circulation, although the role of sheep in EHDV epidemiology seems to be negligible [108]. Other species that have also been seropositive are fallow deer, wapiti, bison, goat, red deer, and roe deer [109].

After blood feeding from an infected animal, EHDV first replicates in the midgut epithelium of the biting midge and disseminates through the hemolymph to secondary infection sites including the salivary glands [110], where viral particles can undergo changes related to infectivity [111]. When an infected *Culicoides* bites a susceptible ruminant host, the virus primarily infects dendritic cells and macrophages. These subsequently migrate carrying the virus to regional lymph nodes, draining the entry site. Here, the primary replication occurs, and, then, the virus is disseminated to many organs, particularly the spleen and lungs (although the virus has also been detected in the heart, cerebrum, cerebellum, and testes). Mononuclear phagocytes and endothelium cells are mainly infected [112]. EHDV replicates in the vascular epithelium, benefiting from autophagy and inducing cell death by apoptosis [112,113,114], leading to hemorrhage and thrombosis. In addition to direct viral damage caused by replication in the endothelium, replication in macrophages and endothelial cells leads to the release of pro-inflammatory cytokines such as interleukin 1 (IL-1) and interleukin 6 (IL-6) [115], which could enhance viral pathogenesis. Thus, inflammatory mediators may contribute to the severity of clinical disease, as well as the induction of vasoactive mediators, such as nitric oxide [116,117].

During the initial steps of infection, a type I IFN response determines whether the disease makes progress in the infected host. In experimental infection of deer with EHDV, peak viraemia coincided with peak IFN type I levels and both then rapidly declined [118]. Importantly, host genetics related to the innate immune response probably play a role in disease outcome [119]. Leukopenia and lymphopenia are common features of EHD. Clinical outcomes vary depending on the different forms of disease. The sub-acute form is characterized by development of ulcers in the oral cavity and the gastrointestinal tract [102]. The acute forms portrays a hemorrhagic disease that includes hyperaemia of the conjunctiva and the oral mucosa, pulmonary edema, pleural effusion, and multifocal haemorrhages in a variety of organs affected by vascular damage and coagulopathy [102]. The peracute disease causes fulminant death, probably due to the development of pulmonary vascular injury with subsequent pulmonary oedema, probably assocciated with the cytokine storm [116]. Prolonged infections have also been observed in experimentally infected WTD and cattle, which can be explained by association of EHDV viral particles with invaginations in the erythrocyte membrane [120].

Although infection is usually less severe in cattle than in WTD, outbreaks among bovines have been more virulent in recent years. EHDV epizootics reporting severe clinical disease in cattle were described in Japan [67], Reunion Island [69], and Turkey [37], with more recent outbreaks in Egypt [41], Israel [44,71], and Tunisia [50]. Increased virulence in bovines has been also detected in North America over the past decade [121]. When cattle are infected with EHDV and clinical signs occur, these can include pyrexia, hyperaemia, oral ulcerations, ptyalism, excessive nasal and ocular secretion, lethargy, weakness, lameness, loss of appetite, reduction in the production of milk, and edemas (palpebral and conjunctival mostly), among others [44,45,67,69]. In some cases, animal deaths have been recorded [69]. Importantly, EHDV was frequently present in cases of aborted cattle during some epizootics [53,71,72,122]

## 5. Experimental Animal Models of EHDV

### 5.1. White-Tailed Deer (WTD) and Other Cervid Species

As the global incidence of EHD is constantly increasing, the study of pathogenesis, transmission, and diagnosis as well as the evaluation of vaccine candidates in natural wildlife hosts is a key issue. In this sense, cervids have been used for EHDV study in natural hosts (Table 1). Different species of cervids, including red, fallow, roe, and muntjac deer, were experimentally infected with the New Jersey strain of EHDV-1, which is highly virulent in WTD. In contrast, these animals did not display any signs of disease, although productive infection could be detected [123]. Among the species of cervids susceptible to EHDV infection, WTD stand out as the most affected host of EHDV. Experimental infection of WTD with EHDV followed the first outbreak detected in the USA. WTD inoculated with EHDV-1 displayed severe illness in most cases, resembling the clinical disease observed in nature and displaying gross and histopathological lesions [124,125]. A high mortality among infected WTD was recorded [125]. In general, experimental infection of WTD with EHDV induces acute disease leading to high mortality rates, independent from the age or sex of the infected animals. WTD infected with the reassortant North American EHDV-6 resembled infection and clinical disease caused by the US strains of EHDV-1 and EHDV-2. Researchers also characterized virological parameters of EHDV infection, detecting viraemia from the third day post-inoculation until more than two weeks later (viraemia was detectable for more than seven weeks) in surviving animals, even in presence of high nAb titers. Viral RNA was detected in tissue samples of organs that presented macro- and microscopic lesions [126]. A field isolate of EHDV-7 that caused intense and widespread epizootic in domestic cattle in Israel was also assessed to determine whether WTD was susceptible to infection. This virus strain led to a clinical disease identical to that observed in experimental infections with North American isolates, and virological parameters were similar to those of EHDV-6 infected WTD [127]. The fact that “exotic” strains of EHDV can productively infect WTD inducing fatal clinical disease highlights the marked susceptibility of WTD to EHDV infection. Nonetheless, it is important to note that not all subspecies of WTD share the same susceptibility grade, with some of them, such as the subspecies *Odocoileus virginianus texanus* (which inhabits regions where EDHV has been endemic for a long period), showing innate resistance to EHDV [128]. Differential expression of pro-inflammatory cytokines could explain this resistance [115]. Host genetic factors can also influence susceptibility to EHDV infection [119]. This illustrates the complexity of understanding the pathogenesis and virulence of EHDV even in WTD. 

The employment of WTD as an EHDV animal model has not been restricted to the characterization of EHDV infection and clinical disease, but has also been used in the study of virus pathogenesis and host defense mechanisms triggered against this pathogen [115,129]. Peak levels of IFN type I (IFN-α and IFN-β) coinciding with peak viraemia levels in infected WTD reflected the progression of an innate immune response following EHDV-1 inoculation. IFNs type I were not detected in blood after a second inoculation of convalescent animals with EHDV-1, due to an abrogated viral replication caused by a neutralizing humoral response [118]. The induction of homologous nAbs is a very common feature observed in surviving WTD and could be explored as a correlate of protection. Some authors have characterized the dynamics of passive immunity against EHDV, observing a prolonged persistence of maternal nAbs and protection against EHDV in fawns [130,131]. Cell-mediated immune responses and suppression of T-cell proliferation have also been recorded in EHDV-infected WTD [103,132]. 

Several works have outlined the utility of WTD as an experimental model for studying EHDV transmission and propagation via the *Culicoides* insect vector. This animal model has been useful in studies of vector competence [133,134] and to determine factors affecting viral replication in infected midges [135]. Interestingly, significant epidemiological features of EHDV were evinced after blood-feeding of *Culicoides* midges from experimentally infected WTD. Mendiola et al. (2019) demonstrated that insect vectors can become infected even when virus cannot be detected in the blood of infected animals [136]. Alternative routes of transmission in WTD have been evaluated. Indeed, reports of oral infection of WTD exist, and oral and fecal shedding of EHDV by infected WTD have been demonstrated, which means that direct and indirect contact transmission may play an important role in farm animals stocked at high densities [137].
microorganisms-11-01339-t001_Table 1Table 1Cervid species used as experimental animal models for study of EHDV infection and pathology.SpecieAgeChallenge VirusDoseInoculation RouteMortalityClinical DiseasePathological AbnormalitiesViremiaVirus Isolation/RNA in TissuesGross and Histopathological LessionsImmune ResponseRefRed deer (*Cervus elaphus*)1–4 years oldEHDV-1USA1955/0110^6^ TCID_50_IVNoNoNoYes ^a^Not evaluatedNot evaluatedHomologous nAbs[123]Fallow deer (*Dama dama*)1–4 years oldEHDV-1USA1955/0110^6^ TCID_50_IVNoNoNoYes ^a^Not evaluatedNot evaluatedHomologous nAbs[123]Roe deer (*Capreolus capreolus*)1–4 years oldEHDV-1USA1955/0110^6^ TCID_50_IVNoNoNoYes ^a^Not evaluatedNot evaluatedHomologous nAbs[123]Muntjac deer (*Muntiacus muntjac*)1–4 years oldEHDV-1USA1955/0110^6^ TCID_50_IVNoNoNoYes ^a^Not evaluatedNot evaluatedHomologous nAbs[123]White-Tailed deer (*Odocoileus virginianus*)2–24 months oldEHDV-1USA1955/01Unknown (inocula from spleen, liver, lung, blood, and kidney of severely infected WTD)SC or IMYes (>56% mortality rate)Severe to fatal clinical diseaseNot evaluatedNot evaluatedNot evaluatedYes Homologous nAbs[125]EHDV isolated inSouth Dakota (no data specified)Yes (>33% mortality rate)9–36 months oldEHDV-1USA1955/01Unknown (inoculum from severely infected WTD spleen)IMYes (>50% mortality rate)YesThrombocytopeniaNot evaluatedNot evaluatedYesNot evaluated[124]5 months oldEHDV-6 (EHDV-6/EHDV-2 reassortant isolate CC-304-06 Indiana, US, 2006)10^6.4^ TCID_50_SC and IDYes (60% mortality rate)YesLeukopenia, lymphopenia, hypoproteinemiaYes ^a^Spleen, lung, lymph node and skinYesHomologous nAbs[126]8 months oldEHDV-7 ISR2006/0410^5.27^ TCID_50_ (inoculum from severely infected WTD blood)SC and IDYes (66.7% mortality rate)YesLeukopenia, lymphopenia, hypoproteinemia, thrombocytopeniaYes ^a^Spleen, lung, lymph node, heart, cerebellum, cerebrum and skinYesHomologous nAbs[127]2-months-old *Odocoileus virginianus texanus*
EHDV-1 (from field isolate from Walton County, Georgia in 1999)10^7^ TCID_50_ (inoculum from severely infected WTD blood)SC and IDNoMild clinical diseaseNoYes ^a^Not evaluatedNot evaluatedHomologous nAbs[128]EHDV-2 (field isolate from Clarke County, Georgia)10^7.1^ TCID_50_ (inoculum from severely infected WTD blood)No2-months-old *Odocoileus virginianus borealis*EHDV-1 (from field isolate from Walton County, Georgia in 1999)10^7.1^ TCID_50_ (inoculum from severely infected WTD blood)Yes (100% mortality rate)Severe clinical diseaseLymphopenia, hypoproteinemiaGross lesions (not specified)EHDV-2 (field isolate from Clarke County, Georgia)10^7^ TCID_50_ (inoculum from severely infected WTD blood)Yes (20% mortality rate)3–4 months oldEHDV-2 (field isolate from Clarke County, Georgia in 1990)10^5.5^ TCID_50_ (inoculum from severely infected WTD blood)SC and IDYes (31% mortality rate)YesLymphopeniaYes ^a^Not evaluatedNot evaluatedHomologous nAbs[118]3–4-months-old EHDV-2 convalescent animalsEHDV-2 (field isolate from Clarke County, Georgia in 1990)10^3^ TCID_50_ (inoculum from severely infected WTD blood)NoNoNoNo ^a^Not evaluatedNo27–47-days-old fawns (feed with colostrum with maternal nAbs)EHDV-2 (field isolate from Clarke County, Georgia in 2016)10^5.6^ TCID_50_ (inoculum from virus isolated in severely infected WTD spleen and propagated in BHK_21_ cells)SC and IDNoMild or absent clinical diseaseNot evaluatedNo (transient viraemia in two fawns ^a^)SpleenNoMaternal homologous nAbs[131]27–47-days-old fawnsNoModerate clinical diseaseYes ^a^3–4 months oldEHDV-2 (strain not specified)10^5.5^ TCID_50_ (inoculum from infected WTD blood)SC and IDYes (25% mortality rate)YesLymphopeniaYes ^a^Not evaluatedNot evaluatedHomologous nAbs[132]4–5 months oldEHDV-2 (from field isolate from Clarke County, Georgia in 2016)10^6.6^ TCID_50_ (inoculum from infected WTD blood)SC and IDNoModerate clinical diseaseNot evaluatedYes ^a^Not evaluatedNot evaluatedHomologous nAbs and Cell-mediated response[103]EHDV-1 (from field isolate from Walton County, Georgia in 1999)10^7.6^ TCID_50_ (inoculum from virus isolated in severely infected WTD spleen and propagated in BHK_21_ cells)Yes (100% mortality rate)Severe clinical diseaseLymphopenia,thrombocytopenia4–5-months-old EHDV-2 convalescent animalsEHDV-1 (from field isolate from Walton County, Georgia in 1999)10^7.6^ TCID_50_ (inoculum from virus isolated in severely infected WTD spleen and propagated in BHK_21_ cells)NoMild or absent clinical diseaseNo4 months oldEHDV-2 (from field isolate from Clarke County, Georgia in 1990)10^5^ TCID_50_ (inoculum from infected WTD blood)SC and IDNot specifiedNot evaluatedNot evaluatedYes ^a^Not evaluatedNot evaluatedNot evaluated[134]6-months-old *Odocoileus virginianus borealis*EHDV-2 (not specified)10^7.03^ TCID_50_Not specifiedYes (25% mortality rate)Moderate or severe clinical diseaseNot specifiedYes ^a^Spleen, lung, buccal mucosa and skinYesInduction of pro-inflammatory cytokines [115]6-months-old *Odocoileus virginianus texanus*Mild or absent clinical diseaseNo8 months oldEHDV-7 ISR2006/04UnknownInfected *C. sonorensis*Yes *Severe clinical diseaseLeukopenia, lymphopenia, hypoproteinemia, thrombocytopeniaYes ^a^Cerebrum, cerebellum, heart, lung, spleen, lymph node, skin, epididymisYesHomologous nAbs[133]4–7 months oldEHDV-7 ISR2006/0410^5.1^ TCID_50_ (inoculum from virus isolated in severely infected WTD spleen and propagated in BHK_21_ cells)SC and IDYes *Not evaluatedNot evaluatedYes ^a^Not evaluatedNot evaluatedNot evaluated[135]7 months oldEHDV-2 (from field isolate from Coffey County, Kansas in 2012)10^6.5^ TCID_50_ (inoculum from virus isolated in infected WTD spleen and propagated in CPAE, BHK_21_, and CuVaW8A cells)SC and IDNot specified *Mild to moderate clinical diseaseNot evaluatedYes ^a^Not evaluatedNot evaluatedNot evaluated[136]4–6 months oldEHDV-1 (from field isolate from Walton County, Georgia in 1999)10^7.1^ TCID_50_ (inoculum from virus isolated in severely infected WTD spleen and propagated in BHK_21_ cells)SC and IDNoMild to severe clinical diseaseNot evaluatedYes ^a^Virus isolated from rectal and oral cavitiesNot evaluatedNot evaluated[137]Inoculation: IM, Intramuscular; ID, Intradermal; SC, Subcutaneous; IV, Intravenous; IC, Intracerebral; IP, Intraperitoneal. * Just one animal was included in the experimental design. ^a^ Viraemia measured by plaque assay.


### 5.2. Cattle and Other Farm Animals

WTD and other wildlife experimental models involve limitations (reviewed in [138]) that are not shared by suitable traditional livestock experimental animals widely used in infectious disease research. As stated above, during recent EHDV outbreaks in North America, the Mediterranean Basin, and Reunion Island, an apparent increase in pathogenicity of EHDV in cattle raised concerns. Therefore, cattle can be considered as a more accessible and cheaper alternative to WTD for studying viral pathology and evaluating vaccine efficacy. In Table 2, we compile data on all the experimental infections carried out in domestic animals other than WTD so far. The first experimental inoculations of cattle with EHDV (Ibaraki-5 and Kyushu-1 strains) date from 1969, inducing clinical disease [67]. In later studies, cattle, sheep, pigs, and goats were inoculated with the New Jersey strain of EHDV-1, which was virulent in deer [123]. Despite technical limitations concomitant with that time, researchers were able to observe viraemia in inoculated sheep and cattle while none of the goats or pigs were viraemic. Interestingly, the virus was recovered from the vulva of a recently lambed viraemic sheep. Clinical disease was not observed in any animal [123]. Similarly, other researchers observed that inoculation of cattle with the EHDV-1 New Jersey strain did not induce clinical disease, but the virus could be isolated from day 9 to day 23 post-inoculation and viraemia was detected by gel-based reverse transcriptase-PCR between days 3 and 28 post-infection. Similar results were observed for cattle infected with the Alberta strain of EHDV-2 [139]. In another work, subsequent inoculation of cattle with two US isolates led to transient viraemia as well as the induction of a neutralizing immune response in absence of clinical disease [140]. Likewise, no clinical disease nor other EHDV-associated abnormalities were observed after inoculation of calves with a virulent EHDV-2 isolate from deer. However, animals did display prolonged viraemia even in the presence of homologous nAbs. Importantly, inoculation of a deer with the same virus did induce severe clinical disease and the animal was euthanized, which indicates a non-virulent phenotype of the inoculum in cattle [141]. Other strains different from US isolates have been used for experimental infection of cattle. Virus isolates from the EHDV-6 outbreaks in Morocco and Turkey in 2006 and 2007, which were characterized by a pathogenic profile in infected bovines, were assessed. As occurred with the US isolates, inoculation of cattle with EHDV-6 TUR2007/01 or EHDV-6 MOR2006/17 strains did not produce any evidence of clinical disease, but viraemia was observed throughout the experiment and viral RNA was detected in the spleens and lymph nodes of infected animals during this viraemia peak. Viraemia was prolonged for more than four weeks and the virus could be isolated from blood three weeks post-inoculation, even when animals seroconverted and presented nAbs [142]. Nonetheless, experimental infection of cattle with the EHDV-6 Reunion Island strain (which elicited clinical signs in cattle in the field) did induce mild clinical signs of disease, although virologic parameters were similar to those of experimental infections with EHDV-6 TUR2007/01 or EHDV-6 MOR2006/17 [143]. Schbaumer and colleagues explored whether previous exposure to BTV or EHDV could influence subsequent infection with the other [144]. Experimental infections of sheep and cattle with a EHDV-7 strain pathogenic in cattle were conducted. No productive infection occurred in inoculated sheep, although two of them were slightly positive according to RT-qPCR at some point after infection. Viraemia was detected in infected cattle in absence of clinical signs of disease, showing similar kinetics to previous observations for EHDV-6 [142]. Moreover, the authors did not find any sign of interference between these two orbivirus infections [142]. A recent work by Sailleau and colleagues reported experimental infection of calves with strains of all EHDV serotypes. Interestingly, while subcutaneous inoculation of EHDV-1, -2, -4, -5, -6, and -8 elicited similar results in terms of clinical signs and virological parameters to those observed in the works previously depicted here, infection with EHDV-7 produced clinical disease in the infected animal, which displayed apathy, diarrhea, prostration, inability to stand, and anorexia, and had to be euthanized in consequence [145]. Importantly, researchers also revealed cross-neutralizing relationships among serotypes, data that can be of great importance in terms of vaccine development.

The incapacity of EHDV isolates to induce the clinical signs observed in the field could rely on an attenuation due to passage in cell culture (although some experimental infections involved inoculums from severely infected deer), or aspects related to the inoculated animal (age, genetic background). The fact that salivary proteins from Culicoides vector are not involved during the described experimental animal inoculations could be important for virus pathogenesis [146]. However, different routes of inoculation as well as virus inoculums were tested by Ruder et al. (2015) and neither clinical disease or differences in virus kinetics were observed when calves were infected with EHDV-7 by subcutaneous, intradermal and/or intravenous inoculation, or by infected Culicoides biting midges [147]. Overall, cattle can be used as a reliable EHDV animal model although further research is needed to elucidate the mechanisms behind the diminished pathogenicity of EHDV in bovine experimental infections. Besides, all these works resemble the ability of EHDV to productively infect European breeds of dairy cattle, which reflects a big concern regarding the potential economic impact of EHDV that is circulating in Europe.

### 5.3. Mouse Models

Availability of appropriate laboratory animal models is a major concern when studying disease pathogenesis and developing efficient and safe therapies against viral diseases. Mouse models are a reliable tool that reproduce or, at least, partially mimic the disease pathogenesis in a variety of cases [148]. For vaccine development, utilization of valid mouse models endorses the basis of every traditional vaccine development procedure as it implies several advantages, such as reduction of costs and time, facility to handle and accessibility of a huge number of optimal reagents. When considering vaccine evaluation against veterinary diseases, mouse models offer a more accessible and affordable animal housing compared to natural hosts. Immunocompromised mouse models, like mice deficient in the type I IFN (IFN-α/β) receptor (IFNAR(−/−)), have been extensively used for vaccine efficacy assessment. The IFNAR(−/−) knock-out receptor mouse model has been employed to study viral infection, disease, pathogenesis and vaccine testing against a plethora of viral diseases [148]. This laboratory animal model has been exploited for extensive study of BTV and AHSV [149,150,151]. Infection of IFNAR(−/−) mice with BTV and AHSV reproduces the pathology observed in natural hosts of these viruses. Moreover, establishment of IFNAR(−/−) mice as a BTV and AHSV infection model has allowed preclinical assays of novel vaccine candidates as a prior step to their evaluation in natural hosts [152,153,154,155]. For EHDV, suckling Swiss mice were intracerebrally inoculated with the fully virulent deer New Jersey strain of EHDV-1, showing illness and being sacrifice in consequence, or succumbing to infection in some cases. Besides, after serial passage through the brains of newborn mice, the authors observed attenuation of this virulent strain in deer [156,157]. However, newborn mice are incompatible with efficacy evaluation of novel vaccine candidates owing to its immature immune system [158]. Much later, Schbaumer, M. et al. (2012) demonstrated that the IFNAR(−/−) mouse model is a suitable small animal model for EHDV [159] (Table 2). Although this study is limited by the low number of animals involved, they observed dose-dependent susceptibility of intraperitoneally (a route of inoculation less complex than the intracerebral one) EHDV-7 inoculated mice, with the more severe affected mice displaying BTV-like clinical signs (except conjunctivitis) and detecting presence of RNA in spleen and gross lesions in spleen and liver of death mice. Thereafter, the IFNAR(−/−) mouse model was used for virus isolation from blood of EHDV-7 infected cattle, observing similar results to those by Schbaumer, M. et al. (2012) [144,159]. Despite these results are very promising in order to explore the possibility of establishing the well-known IFNAR(−/−) mice as an infection model for EHDV, no further research has been conducted in this sense. As the global incidence of EHDV is increasing, assessment of virulence and pathogenesis of other EHDV isolates in the IFNAR(−/−) mouse model should be explored in the future.
microorganisms-11-01339-t002_Table 2Table 2Non-cervid experimental animal models used for study of EHDV infection and pathology.SpeciesBreedChallenge VirusDoseInoculation RouteMortalityClinical DiseaseViremiaRNA in TissuesTissue DamageImmune ResponseRefCattleJapanese blackKyushu-1 and Ibaraki-5 strains1-5 mL infected bloodIVNoYes (conjunctivae and oro-nasal inflammation, fever, leukopenia) Not evaluatedNot evaluatedNecrotic changes of epithelial cellsNot evaluated[67]6–18-months-old Jersey or Friesian cattleEHDV-1USA1955/0110^6^ TCID_50_IVNoNoYes ^a^Not evaluatedNot evaluatedHomologous nAbs[123]4–6-months-old calvesEHDV-1USA1955/0110^6^ TCID_50_ID and SCNoNoYes ^a,b^Not evaluatedNot evaluatedHumoral response to EHDV-1 or EHDV-2 (neutralization not tested)[139]EHDV-2CAN1962/017–9-months-old Holstein-Friesian cattleEHDV-6 TUR2007/0110^7.5^ TCID_50_SCNoNoYes ^a,c^Spleen and lymph nodesNoHomologous nAbs
EHDV-6 MOR2006/1710^7.5^ TCID_50_12-months-old Holstein cattleEHDV-7/ISR2006/135 × 10^5^ TCID_50_SCNoNoYes ^a,c^Not evaluatedNot evaluatedHomologous nAbs[144]18-months-old Holstein cattleEHDV-6 (Reunión Island, 2008)Two doses of 6 × 10^6^ TCID_50_SCNoYes (conjunctivitis, epiphora.One animal displayed moderate oedema, gingival ulcer)Yes ^a,c^Spleen, liver, skin, kidney, lymph nodes and heartNot evaluatedHomologous nAbs[143]1-year-old Holstein steer and adult Holstein cattleEHDV-6 (EHDV-6/EHDV-2 reassortant isolate CC-304-06 Indiana, US, 2006)10^7.27^ TCID_50_SC and IDNoNoYes ^a^NoNoHomologous nAbs[126]>4-years-old Holstein cattleEHDV-7 ISR2006/04 (WTD blood inoculum)10^6.1^ TCID_50_
ID and SCNoNoYes ^c^Spleen and lymph nodeNoHomologous nAbs[147]2-month-old Holstein calvesEHDV-7 ISR2006/04 (BHK cell culture supernatant)10^7.12^ TCID_50_ID and SCNoNoYes ^c^Spleen, lung and lymph nodeNoHomologous nAbsID, IV and SCNoNo (elevatedrectal temperature at two timepoints)Yes ^c^Spleen, lung and lymph nodeNoHomologous nAbsEHDV-7 ISR2006/04 UndeterminedInfected *C. sonorensis*NoNoYes ^c^Spleen, lung and lymph nodeNoHomologous nAbs2-month-old Prim’Holstein calvesEHDV-1 USA1955/0110^7^ TCID_50_ *I.M.NoNoYes ^c^Not evaluatedNot evaluatedHomologous nAbs. Low nAb titers against EHDV-7 ISR2006/01 [145]EHDV-2CAN1962/015 × 10^7^ TCID_50_ *NoNoHomologous nAbs. Low nAb titers against EHDV-1 USA1955/01 and EHDV-7 ISR2006/01 EHDV-4NIG1968/013 × 10^6^ TCID_50_ *NoNoHomologous nAbs. Low nAb titers against EHDV-5 AUS1979/06EHDV-5 AUS1979/061.6 × 10^6^ TCID_50_ *NoNoHomologous nAbs. Low nAb titers against EHDV-4 NIG1968/01EHDV-6AUS1981/0710^7^ TCID_50_ *NoNoHomologous nAbs. EHDV-7ISR2006/0110^7^ TCID_50_ *Yes (euthanasia)Yes (apathy, diarrhea, prostration, unable to stand, anorexia)Homologous nAbs. Low nAb titers against EHDV-2 CAN1962/01EHDV-8AUS1982/065 × 10^7^ TCID_50_ *NoNoHomologous nAbs. Low nAb titers against EHDV-6 AUS1981/07Sheep12–24-months-old Suffolk cross or Dorset Horn sheepEHDV-1USA1955/0110^6^ TCID_50_IVNoNoYes ^a^Not evaluated(Presence of virus in vulvae of a viraemic sheepNot evaluatedHomologous nAbs[123]Cheviot-SouthDown sheep (unknow age)EHDV-1USA1955/01Unknown (inoculum from severely infected WTD spleen)IMNoNoNot evaluatedNot evaluatedNoNot evaluated[124]10-months-old East Frisian sheepEHDV-7/ISR2006/135 × 10^5^ TCID_50_SCNoNoNoNot evaluatedNot evaluatedNot evaluated[144]Goat6–24-months-old British Alpine goatEHDV-1USA1955/0110^6^ TCID_50_IVNoNoNoNot evaluatedNot evaluatedHomologous nAbs[123]Pig3-months-old Large WhitesEHDV-1USA1955/0110^6^ TCID_50_IVNoNoNoNot evaluatedNot evaluatedHomologous nAbs[123]MouseSwiss newborn miceEHDV-1USA1955/01Not specifiedICYes (98.6% mortality rate, 436 out of 442 mice)Yes (loss of postural reflexes, irregular respiration, cyanosis, and tonic and clonic convulsions)Not evaluatedNot evaluatedNot evaluatedNot evaluated[156,157]Adult IFNAR(−/−) miceEHDV-7/ISR2006/13Unknown (blood inoculum from infected cattle)IPYes (8.33% mortality rate, 1 out of 12 mice)Yes (ruffled fur, apathy (no conjunctivitis))Yes ^c^Spleen (in 10 out of 12 mice)Enlarged spleensNot evaluated[144]Adult IFNAR(−/−) miceEHDV-7/ISR2006/135 × 10^2^ TCID_50_IPYes (30% mortality rate, 1 out of 3 mice)Yes (ruffled fur, apathy (no conjunctivitis))Not evaluatedSpleen (in both dead and surviving mice)Enlarged spleens and necrotic foci in liver in dead animals)Not evaluated[159]5 × 10^5^ TCID_50_Yes (100% mortality rate, 2 out of 2 mice)Inoculation: IM, intramuscular; ID, intradermal; SC, subcutaneous; IV, intravenous; IC, intracerebral; IP, intraperitoneal. * Inoculation dose could vary not completely specified; ^a^ viraemia measured by plaque assay; ^b^ viraemia detected by gel-based reverse transcriptase-PCR; ^c^ viraemia measured by real-time-qPCR.


## 6. Classic and Novel Vaccine Approaches against EHDV

Vaccination entails the most effective countermeasure to successfully contain several human and veterinary viral diseases. In the case of EHDV, vaccines based on conventional approaches have been developed and their commercialization has been circumscribed to regions where the virus has circulated causing a significant economic impact. In Japan, two vaccines against EHDV-2 are commercially available: a monovalent live attenuated vaccine and an inactivated bivalent vaccine (against EHDV-2 and bovine ephemeral fever, caused by bovine ephemeral fever virus). The monovalent live attenuated vaccine of serotype 2 has been demonstrated highly immunogenic and relatively safe, as the virus was isolated from a vaccinated animal in a recent study [160]. In the USA, where serotypes 1 and 6 of EHDV are endemic and cause recurrent outbreaks, autogenous vaccines have been commonly used. However, no peer-reviewed data exist regarding the protection efficacy of these inactivated vaccines, which could also imply important issues regarding animal welfare [161]. There is no currently licensed EHDV vaccine in Europe as there has not yet been a real need for it. However, considering the wide expansion of EHDV-6 and EHDV-8 in the Mediterranean Basin and the recent arrival of EHDV-8 in Europe, as well as the epidemiological history of EHDV, closely related to BTV, the development and commercialization of effective vaccines against EHDV is needed.

LAVs (live attenuated vaccines) against BTV are used in the United States, Turkey, the Republic of South Africa, India, and Israel, among others [161]. However, BTV LAVs, which show highly immunogenicity, are often associated with several drawbacks relating toanimal welfare and transmission to insect vectors (reviewed elsewhere [161]). For these reasons, and after being used to control several outbreaks of BTV over the years, immunization with BTV LAVs was reduced and, eventually, completely substituted by inactivated vaccines in the European Union (EU) [162]. Therefore, LAVs are not a recommended choice to consider in EHDV vaccination campaigns in the EU, as the possibility of virus spillover to unaffected regions through uptake and spread by midges or in-contact transmission is clear. Inactivated vaccines against BTV are produced and licensed in Europe, and, although some pitfalls exist (reviewed in [162]), this approach has demonstrated more than enough efficacy to control this disease [162]. Once inactivated vaccines are positively tested and licensed, EU authorities should make a deep analysis based on a risk assessment to determine whether implementation of mass vaccination in the affected territories (southern Spain and Italian islands) is convenient. The experience of mass vaccination against BTV in affected and risk areas of Europe has been worthwhile. BTV circulation and occurrence of clinical disease disappeared in regions where vaccination reached 80%, limiting the estimated economic losses [163]. 

Next-generation vaccines against EHDV must overcome inherent disadvantages of classical approaches that occur with BTV and AHSV. First, they must allow differentiation between naturally infected and vaccinated animals (DIVA), which has fundamental implications in the economic field. Second, these newly generated vaccines should induce protection against multiple EHDV serotypes, whose expansion to non-endemic latitudes is highly probable. To date, the unique vaccine candidate that has been evaluated is based on recombinant VP2 protein of EHDV-2 [164] (Table 3). This DIVA subunit vaccine has shown promising results in terms of immunogenicity and protection in the primarily EHDV affected host, WTD, preventing it from EHDV clinical disease, infection, and viraemia. Prime-boost immunization with rVP2 of serotype 2 induced high titers (ranging from 1:240 to 1:320) of homologous nAbs in immunized WTD. After viral challenge with virulent EHDV-2, immunized animals did not display EHD-related signs of disease and showed steady rectal temperatures and peripheral lymphocyte counts. No viraemia nor RNA were detected in EHDV-target organs, and immunized WTD showed an absence of gross and histopathological lesions. Although not yet evaluated, its efficacy for avoiding transmission to the insect vector seems plausible as no RNA was detected in blood. This EHDV-2 rVP2-based vaccine is currently under field trial in the USA. Importantly, cattle immunized following the same immunization strategy also developed a potent humoral response. Not only that, but the authors also achieved the expression and purification of the rVP2 of EHDV-6, which induced high titers of homologous nAbs in cattle. In this sense, bi- or multivalent vaccines could be formulated as proposed in the study [164]. Different vaccine platforms widely used for novel BTV and AHSV vaccines [152,165] should be applied for generation of novel EHDV vaccines, e.g., subunit vaccine and viral vector-based vaccines. In this regard, Alshaikhahmed and Roy (2013) exploited their experience with non-infectious BTV-VLPs to develop core-like particles (CLPs, composed of VP3 and VP7) and virus-like particles (VLPs, composed of VP3, VP7, VP5 and VP2) of EHDV-2 [166] (Table 3). Immunization with two doses of safe DIVA VLPs induced nAbs against EHDV-2 in rabbits. Low titers of heterologous nAbs against EHDV-2 and -6 were detected but this was probably due to the animal model used as for BTV [167]. No immunogenicity nor protection were assessed in EHDV natural hosts. Importantly, authors demonstrated that EHDV-2 CLPs can served as scaffold for rapid generation of VLPs of different serotypes, which opens the opportunity for generating cocktails of EHDV VLPs. Despite this same strategy providing good results in sheep against BTV, these vaccine candidates have not yet been commercialized, which may be due to low affordability. This could constrain future implementation of EHDV VLPs. Plant-based generation of EHDV VLPs offers a possible solution to this problem. Other authors also generated VLPs of EHDV-6, but data on immunogenicity or protection are lacking [168] (Table 3). Another plausible innovative vaccination approach is the generation of non-replicative live-attenuated EHDV vaccines, as reverse genetics (RG) systems have been successfully implemented for EHDV [169,170]. Development of this technology for BTV and AHSV allowed the production of the innovative disabled infectious single cycle (DISC) and disabled infectious single animal (DISA) vaccines [171,172]. DISA and DISC vaccines are DIVA [173,174,175], effective, and completely safe candidates and, although serotype-specific, cocktails of different serotypes of disabled viruses provide high immunogenicity and multiserotype protection in sheep and cattle [176,177,178].

Eventually, the first cases of EHDV-8 detected within the European Union will drive research on next-generation EHDV vaccines. However, the unpredictability of EHDV epidemiology forces researchers to explore multiserotype vaccine approaches. Infection- and vaccine-induced immune responses against orbiviruses include neutralizing (directed against the VP2 protein) and non-neutralizing humoral as well as cytotoxic cellular immune responses directed toward other structural and nonstructural viral proteins. However, little is known about the antigenicity of the different structural and nonstructural proteins of EHDV. Considering the experience gathered from the generation and study of novel vaccines against BTV and AHSV, further research on host responses against EHDV is warranted. Experimental infection of WTD with EHDV-2 induced cross-protection against EHDV-1 in the absence of EHDV-1 nAbs, indicating the trigger of an EHDV-specific cell-mediated immune response. Indeed, animals were prevented from developing clinical disease after EHDV-1 inoculation, but viraemia was detected. A very similar outcome was observed after induction of protective NS1 and NS2-Nt-specific cell responses in sheep challenged with virulent BTV-4 [155]. A recent article highlighted the existence of shared B- and T-cell epitopes between the sequence of the EHDV and BTV VP7 and VP5 proteins, by in silico analysis [179]. Further, monoclonal antibodies directed against VP7 of EHDV also reacted against BTV VP7, which highlights the presence of discontinuous epitopes shared between these two orbiviruses [180]. Nonetheless, without dismissing this approach, experimental infection of EHDV-2 convalescent WTD with BTV-10, or infection of BTV convalescent cattle with EHDV-7, proved that previous exposure to one orbivirus did not impair viral replication of the other one [118,144]. Further research should be carried out to evaluate the cross-reactivity and potential cross-protection induced by EHDV/BTV antigens.
microorganisms-11-01339-t003_Table 3Table 3Experimental vaccine candidates designed against EHDV.Vaccine TypeAntigen IncludedSerotype of AntigenAnimal ModelDoseAdjuvantImmunogenicityChallengeProtectionRefRecombinant VP2 protein ^1^VP2EHDV-6 (Indiana 2012, 12-38993)6-weeks-old female CD1-ISGTwo doses of 20 μgMontanide ISA25 adjuvantNeutralization against EHDV-6Not challenged-[164]3-4-months-old male HolsteinTwo doses of 150 μgNeutralization against EHDV-6VP2EHDV-2 (Alberta 1962, SV-124-Canada)6-weeks-old female CD1-ISGTwo doses of 20 μgMontanide ISA25 adjuvantNeutralization against EHDV-24-months-old male Holstein calvesTwo doses of 150 μgNeutralization against EHDV-2
5-month-old male WTDTwo doses of 150 μgNeutralization against EHDV-210^6.74^ PFU EHDV-2 (Kansas 2012, strain cc12-304)Yes ^a,b,c,d,e^CLP ^1^VP7, VP3EHDV-1USA1955/01RabbitThree doses (first dose: 500 µg; second and third doses: 250 µg)Incomplete Fruend’s adjuvantInduction of VP3- and VP7-specific antibodies.Not challenged-[166]VLP ^1^VP2, VP5, VP7, VP3Neutralization against EHDV-1/Low neutralization against EHDV-2 and EHDV-6VLP ^1^VP2, VP5, VP7, VP3VP3 and VP7 from EHDV-1USA1955/01. VP2 and VP5 from EHDV-2CAN1962/01Not evaluated in animal model-----VLP ^1^VP2, VP5, VP7, VP3EHDV-6 (MOR2006/05)Not evaluated in animal model-----[168]^1^ Expressed by recombinant baculovirus expression system: ^a^ steady rectal temperature; ^b^ absence of EHDV-clinical signs; ^c^ absence of lymphopenia; ^d^ absence of RNA-emia and RNA in EHDV-target organs; ^e^ absence of gross and histopathological lesions.


## 7. Conclusions

EHDV, an important arthropod-transmitted RNA virus that infects different wild and domestic ruminants, has experienced a northern spread into novel areas in the last 20 years. Global warming may result in expansion of vector species to previously vector-free regions, and in altered vector competence of some midge species. These factors and others related with human activities are likely to increase the risk of EHDV outbreaks in new territories. Worryingly, the virus has recently been detected for the first time in the European Union in October 2022. 

White-tailed deer are especially susceptible to severe illness caused by EHDV infection; nevertheless, disease can also occur in bovines. The increased virulence of certain EHDV strains observed in cattle and the expansion of competent vectors involved in EHDV transmission make necessary further investigation regarding the development of new diagnostic techniques, safe DIVA vaccines, and the evaluation of laboratory animal models that will facilitate the study of the protective capacity of new vaccine candidates against EHDV. 

## Figures and Tables

**Figure 1 microorganisms-11-01339-f001:**
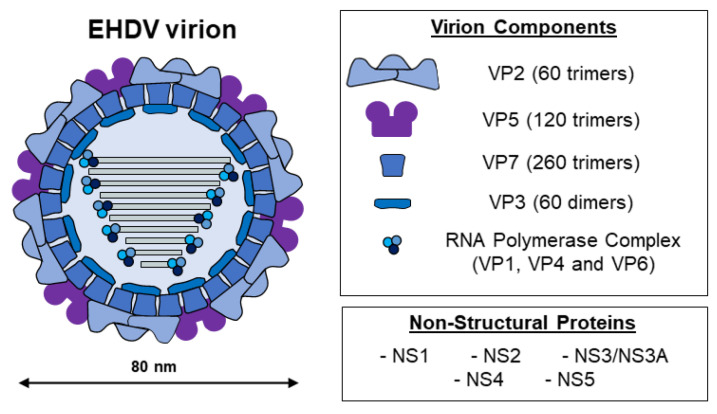
Diagrammatic representation of the viral particle of EHDV (~80 nm in diameter). Three concentric layers constituted by VP2 and VP5 form the outer capsid. The intermediate layer and the subcore are composed by VP7 and VP3. The inner capsid contains the RNA polymerase complex, composed by structural proteins VP1, VP4, and VP6. At least four additional proteins (NS1, NS2, NS3/NS3A, and NS4) are expressed during the replicative cycle.

**Figure 2 microorganisms-11-01339-f002:**
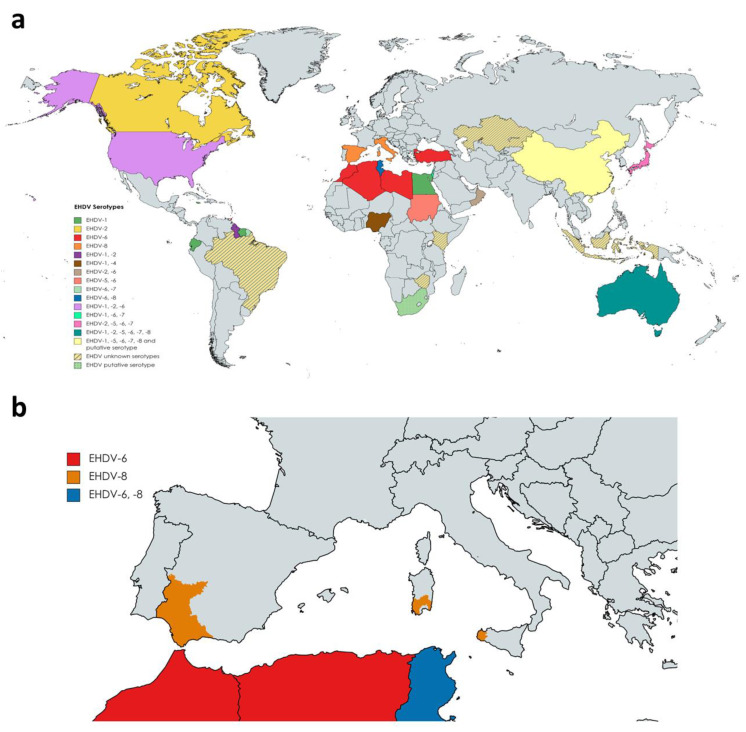
Epidemiology of EHDV across the world (**a**) and in Europe (**b**). (**a**) Colored areas represent the countries were different serotypes of EHDV have caused outbreaks over the years or serological evidence exists. The corresponding serotypes of each country are indicated. Where outbreaks are located within a specific region, the whole country is indicated as infected. (**b**) Representation of the Italian and Spanish affected regions.

## Data Availability

Not applicable.

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
