# Peer review of "Epizootic Hemorrhagic Disease Virus: Current Knowledge and Emerging Perspectives"

_microorganisms, 2023, doi:10.3390/microorganisms11051339_

Round 1

Reviewer 1 Report

General comments

This paper describes a wide range of information on epizootic hemorrhagic disease (EHD), including characteristics and distribution of EHD virus (EHDV), clinical manifestations and pathological features of EHD, and information about animal experiments and vaccine development. Although there have been several review articles about EHD published to date, and some of the information in this paper overlaps with the previous review articles, the information presented in this paper is extensive and up-to-date, and I believe that readers interested in EHD will find this paper valuable for a deeper understanding of EHD. Although this article is excellent in both structure and content, there are a few points that could be considered for revision, which are listed below.

Specific comments

P. 1, lines 14-15, P. 4, lines 155-156, P. 15, line 594

The authors mentioned that the expansion of the geographical distribution of EHDV was mainly due to global climate change, but I do not think there is clear evidence for the idea. Factors influencing the expansion of the EHDV are not limited to the distribution of the vector arthropods, Culicoides biting midges. Livestock movement, susceptibility of Culicoides biting midges and/or host animals to EHDV strains could be the factors and straying of virus-infected midges into airplanes are also possible factors. Although many papers have described the relationship between the ecology of mosquitoes and global climate change to date, the relationship between the ecology of Culicoides biting midges and global climate change is largely unexplored. I agree with the idea that global climate change may be affecting the distribution of arboviruses including EHDV, but given the limited evidence, the description should be like “…expanding its distribution conceivably due to global climate change”.

P. 7, lines 303-304

Golender et al. reported that EHDV have been detected in aborted cases of cattle in Israel, but they did not state that EHDV caused abortion. I think that the wording in lines 303-304 needs to be corrected.

P. 9, “5.2. Cattle and other farm animals”

Since the authors mention the virulence of EHDV in cattle in this paper, I think it would be better to add a reference on the experimental infection using a Japanese strain of EHDV-2 (Ibaraki virus) in cattle below.

Omori et al., Jpn J Microbiol. 13(2):139-157. 1969. Ibaraki virus, an agent of epizootic disease of cattle resembling bluetongue. I. Epidemiologic, clinical and pathologic observations and experimental transmission to calves.

https://pubmed.ncbi.nlm.nih.gov/4309396/

References

Please state authors names correctly. (Ref #58, 59, 71, 112)

Author Response

REVIEWER 1.

We have provided a point-by-point response in red. Hopefully, we have answered your questions properly. Thank you for all the comments.

I agree with the idea that global climate change may be affecting the distribution of arboviruses including EHDV, but given the limited evidence, the description should be like “…expanding its distribution conceivably due to global climate change”. 

We understand the point of view of the reviewer and agree with the idea that other factors can influence the distribution of EHDV. Therefore, we have changed the sentences in

L 15: “…expanding its distribution conceivably due to global climate change”

L155: This is likely to be related to global warming

L:594. Conclusion has been modified.

  1. 7, lines 303-304

Golender et al. reported that EHDV have been detected in aborted cases of cattle in Israel, but they did not state that EHDV caused abortion. I think that the wording in lines 303-304 needs to be corrected.

The sentence has been changed as follow: Importantly, EHDV was frequently present in aborted cases of cattle during some epizootics [71, 72,122].

  1. 9, “5.2. Cattle and other farm animals”

Since the authors mention the virulence of EHDV in cattle in this paper, I think it would be better to add a reference on the experimental infection using a Japanese strain of EHDV-2 (Ibaraki virus) in cattle below.

Omori et al., Jpn J Microbiol. 13(2):139-157. 1969. Ibaraki virus, an agent of epizootic disease of cattle resembling bluetongue. I. Epidemiologic, clinical and pathologic observations and experimental transmission to calves. https://pubmed.ncbi.nlm.nih.gov/4309396/

We agree with the reviewer and this reference has been stated in the text: reference number 67.

References: Please state authors names correctly. (Ref #58, 59, 71, 112). We have changed the format of author’s names in these references.

  1. Murota, K.; Ishii, K.; Mekaru, Y.; Araki, M.; Suda, Y.; Shirafuji, H.; Kobayashi, D.; Isawa, H.; Yanase, T.
  2. al-Busaidy S.M. and Mellor P.S.
  3. Golender, N.; Khinich, Y.; Gorohov, A.; Abramovitz, I.; Bumbarov, V.
  4. McLaughlin, B.E.; DeMaula, C.D.; Wilson, W.C.; Boyce, W.M.; MacLachlan, N.J.

Reviewer 2 Report

This is a timely and important manuscript because it combines the most recent work on EHDV vectors and vaccines in the context of the recent outbreaks of EHD in Europe. The authors do a good job setting up the situation by reviewing the virus, epidemiology (importance/relevance), and vectors and then introducing the pathology and models which will then be used to address evaluation of the vaccines (mitigation strategies).  The organization is very well done and the authors summarize the available literature very well. (Minor note the tables and figure are difficult to read when printed. Figure 2 should list the virus serotypes in a logical order such as EHDV1, then EHDV2, etc. Then the locations with two serotypes starting with 1,2, then 1,3, etc. And continue adding additional serotypes. As it reads now the key seems random.)  

Some of the sections are much better written than others and likely all the authors need to review the entire manuscript for grammar.

Minor note: After so much work summarizing the literature, the conclusion probably warrants more than a single run on sentence. Please break it up into ideas that match the main points of your sections to summarize and drive home the conclusion of each section.  

As stated above, please review the English in some of the sections. Most of them are fine but some need minor corrections to fix inappropriate words that don't fit the sentences.

Author Response

REVIEWER 2. Specific comments

Figure 2 should list the virus serotypes in a logical order such as EHDV1, then EHDV2, etc. Then the locations with two serotypes starting with 1,2, then 1,3, etc. And continue adding additional serotypes. As it reads now the key seems random. We have modified the legend of figure 2 in order to list the serotypes as the reviewer said.  

Some of the sections are much better written than others and likely all the authors need to review the entire manuscript for grammar. We have reviewed the whole manuscript, introducing some changes in order to improve the grammar.

Minor note: After so much work summarizing the literature, the conclusion probably warrants more than a single run on sentence. Please break it up into ideas that match the main points of your sections to summarize and drive home the conclusion of each section. 

The conclusion has been changed .

I hope that we have answered your questions properly. Thank you for all the comments.

Reviewer 3 Report

The review manuscript describes the current state of knowledge on EHDV and the disease caused by the virus in different animal species. The focus is on the structure of the virus, the different serotypes, host species, animal models and vaccines. Where appropriate, reference is made to the closely related BTV.

It is a comprehensive overview, but also well summarised for complicated issues such as interactions between serotypes and host species or cross-resistance.

In terms of content, the manuscript seems mature to me, but the linguistic quality still needs to be improved and the expression sharpened in places.

I have only one minor comment concerning the content: L 513-516: Mass vaccination against BTV8 proved successful not only in Italy, but in whole Europe where it was implemented. And I do not understand the sentence with "firewall". Which strategy and why "firewall"?

The following list of points for linguistic revision is not exhaustive, the whole manuscript should be looked at.

There are problems with tenses, plural and singular, references, punctuation and the use of abbreviations.

L 155-156: "warming of global temperatures"

L169: "in this continent".

L 170-175

L232: "BTV has been expanded"

L235:"continues outbreaks"

L241:"Studies has"

L294: better use outbreaks instead of cases; Check word order in same sentence.

L320: "mortality rates...animal" should be animals as it is a rate.

L382:" None of the goat or pig" should be goats or pigs

L414:"Moreover, authors" article missing.

L428: "...no salivary...not found..." double negotiation

L437:"EHDV now that is circulating"

L455:"has enabled to perform"

L499:"LAV" not introduced. I assume live attenuated vaccine

L593: word order

Author Response

Thank you very much for your comments. I hope that we have answered your questions correctly. 

REVIEWER 3:

In terms of content, the manuscript seems mature to me, but the linguistic quality still needs to be improved and the expression sharpened in places. We have reviewed the grammar and tried to improve the linguistic quality.

I have only one minor comment concerning the content: L 513-516: Mass vaccination against BTV8 proved successful not only in Italy, but in whole Europe where it was implemented. And I do not understand the sentence with "firewall". Which strategy and why "firewall"?

This is right; we have changed Italy for Europe, since inactivated vaccines have been employed among European countries.

We have deleted the last sentence of the paragraph, as we considered a bit confusing.

The following list of points for linguistic revision is not exhaustive, the whole manuscript should be looked at.

There are problems with tenses, plural and singular, references, punctuation and the use of abbreviations.

We hope that new changes solve these problems. If there is something else that we can change to improve the grammar, please let us know, we would be willing to make the necessary changes.

L 155-156: "warming of global temperatures". Has been replaced by: “global warming”

L169: "in this continent". Has been replaced by: “on the continent for the first time”.

L 170-175. I am not sure what is wrong in this sentence. I have replaced it by: EHDV was identified as the causative agent of cattle disease outbreaks in Sicily and southwestern Sardinia.

L232: "BTV has been expanded" and L235:"continues outbreaks".

This sentence has been modified: BTV has spread across central and north Europe since appeared in the continent in 2006, leading to continuous outbreaks and great economic losses

L241:"Studies has".  Have

L294: better use outbreaks instead of cases; Check word order in same sentence. Sentence has been modified: Although infection is usually less severe in cattle than in WTD, recent outbreaks among bovines have been more virulent in last years.

L320: "mortality rates...animal" should be animals as it is a rate. Ok.

L382:" None of the goat or pig" should be goats or pigs. Ok.

L414:"Moreover, authors" article missing. Reference included

L428: "...no salivary...not found..." double negotiation. Changed.

L437:"EHDV now that is circulating". "Now" has been deleted

L455:"has enabled to perform". This has been changed: has allowed

L499:"LAV" not introduced. I assume live attenuated vaccine. This has been included.

L593: word order. The sentence has been changed.